# Photon-free (s)CMOS camera characterization for artifact reduction in high- and super-resolution microscopy

Robin Diekmann[1,3], Joran Deschamps[1,4], Yiming Li [1,5], Takahiro Deguchi [1], Aline Tschanz [1,2], Maurice Kahnwald[1,6], Ulf Matti [1,7] & Jonas Ries [1✉]

Modern implementations of widefield fluorescence microscopy often rely on sCMOS cameras, but this camera architecture inherently features pixel-to-pixel variations. Such variations lead to image artifacts and render quantitative image interpretation difficult. Although a variety of algorithmic corrections exists, they require a thorough characterization of the camera, which typically is not easy to access or perform. Here, we developed a fully automated pipeline for camera characterization based solely on thermally generated signal, and implemented it in the popular open-source software Micro-Manager and ImageJ/Fiji. Besides supplying the conventional camera maps of noise, offset and gain, our pipeline also gives access to dark current and thermal noise as functions of the exposure time. This allowed us to avoid structural bias in single-molecule localization microscopy (SMLM), which without correction is substantial even for scientific-grade, cooled cameras. In addition, our approach enables high-quality 3D super-resolution as well as live-cell time-lapse microscopy with cheap, industry-grade cameras. As our approach for camera characterization does not require any user interventions or additional hardware implementations, numerous correction algorithms that rely on camera characterization become easily applicable.

[1] Cell Biology and Biophysics Unit, European Molecular Biology Laboratory (EMBL), Heidelberg, Germany. [2] Collaboration for Joint PhD Degree Between EMBL and Heidelberg University, Faculty of Biosciences, Heidelberg, Germany. [3] Present address: LaVision Biotec GmbH, Bielefeld, Germany. [4] Present address: Fondazione Human Technopole, Milan, Italy. [5] Present address: Department of Biomedical Engineering, Southern University of Science and Technology, Shenzhen, China. [6] Present address: Friedrich Miescher Institute for Biomedical Research, Basel, Switzerland. [7] Present address: Abberior Instruments GmbH, Göttingen, Germany. ✉email: jonas.ries@embl.de

Scientific complementary metal oxide semiconductor (sCMOS) cameras are increasingly popular for scientific imaging including fluorescence and super-resolution microscopy. For quantitative analysis of the images, pixelwise properties of the camera must be well characterized and accounted for in the analysis algorithm to avoid artifacts. This approach has been used to remove camera artifacts in both single molecule localization microscopy (SMLM)[1,2] and diffraction-limited imaging[2–4]. Specific correction software is readily available[1,3–6], but tools which can easily acquire the necessary data for pixel-dependent noise, offset, and photon response are still missing. Additionally, pixels feature individual dark current characteristics[5], rendering both noise and offset functions of the camera exposure time, which is often neglected in characterization and correction algorithms. Consequently, a majority of sCMOS data is analyzed without explicit camera correction[7]. Industry-grade cameras approach the specifications of scientific-grade cameras and are increasingly used in the scientific community[8–15]. Especially for those cameras, a precise characterization and correction of the large pixelwise variability is indispensable.

Here, we developed a fully automated camera characterization pipeline, which determines pixel- and exposure time-dependent noise, offset and gain maps that are the basis for numerous camera correction algorithms. Our pipeline does not require any specific camera illumination, as it relies solely on dark current and associated thermal noise. In addition to gain, offset and noise maps, it also allows for the explicit consideration of dark current and thermal noise in the image reconstruction, which is of particular importance for long exposure times in SMLM or low light level live-cell imaging. We demonstrate that we can accurately characterize diverse (s)CMOS cameras and use the calibrations to avoid bias in 2D and 3D SMLM and in diffraction-limited imaging. Our camera characterization algorithm is implemented for the popular software packages Micro-Manager[16] as well as ImageJ/Fiji[17] and enables (s)CMOS specific corrections for the broad imaging community.

## Results and discussion

**Camera characterization via dark current.** Camera characteristics are conventionally determined by evaluating mean and variance of the signal in each pixel over many images at several light levels[1]. By approximating the normal distributed readout noise $RN_k$ (standard deviation, in the unit of electrons, $k$ denotes the pixel indices) with a Poisson distribution, one can expect the sum of the detected electrons without light (sum of Poisson distributed dark current) and readout noise to approximate a Poisson distribution with a variance of $RN_k{}^2 + TN_k{}^2*t$, where $TN_k$ denotes the noise (standard deviation) introduced from thermal noise in the $k_{th}$ pixel per time (electrons/sqrt(second)). $t$ is the camera exposure time. Thus, the mean and variance of the signal with no light reaching the camera (i.e., t = 0) correspond to offset and read noise squared, respectively. Due to the stochastic and discrete nature of photon detection, the gain can be calculated as the ratio of the variance and mean signal at different light levels. Thermal excitation is an alternative source for excited electrons, resulting in exposure time dependent dark current $DC_k*t$ that increases the offset. Accordingly, a calibration loses its validity when a different exposure time is used for imaging. This holds particularly true for long exposure times or uncooled cameras.

We turn this source of error to our advantage and use thermal excitations to fully characterize the camera without any light reaching the detector. This is possible as thermally excited electrons follow Poisson statistics just as photoelectrons (Fig. 1a,

Supplementary Fig. 1). Photon-free camera characterization is based solely on dark current and thermal noise (Fig. 1b), using the linear relation between exposure time and dark current to generate different signal levels (Fig. 1c). Extrapolation to 0 ms exposure time gives the baseline $BL$ (i.e. the offset free of thermal effects) as well as read noise $RN$ squared (i.e. the noise free of thermal effects). Additionally, the explicit knowledge of the dark current and thermal noise as a function of exposure time now allows for computation of thermal effects at arbitrary exposure times (Fig. 1d). For comparison, we used the traditional approach of varying light levels at a single frame exposure time of 10 ms (Fig. 1e, f). Notably, the increased mean offset of 0.56 counts as compared to the photon free measurement equals the expected average dark current for 10 ms exposure time. For further verification, we calibrated an uncooled CMOS camera twice on the same day and found no considerable difference for all parameters (Supplementary Fig. 2). However, minor changes in camera parameters were observable over years (Supplementary Fig. 3). We then compared the predictions based on our approach to the experimentally directly determined pixel-dependent offset and noise at different exposure times (Fig. 1g, Supplementary Fig. 4). These comparisons showed high similarity, with average relative errors less than 0.4% for the mean pixel values and 1.3% for the noise (Fig. 1g). For the gain estimation (Fig. 1f), we additionally compared our results with the single shot fluorescence method presented by Heintzmann et al.[18] that is based on out-of-band information from diffraction limited fluorescence images. The relative deviation in the median gain by the different methods was below 3.4%. We conclude that our method in determining the relevant camera characteristics is equivalent to the traditional approaches, but offers the advantages of full automation and calculation for arbitrary exposure times. Note that our approach operates in the very low signal regime of a few electrons only, and so, the gain estimation on the single pixel level is not very precise. Therefore, we used the median of all single pixel gain values as one global gain value. To additionally consider variations in sensitivity (e.g. due to differences in quantum efficiency) between neighboring pixels, we optionally added the flat-fielding approach of Lin et al.[5] and multiplied the flatfield map with the median gain to calculate the photon response map. However, for the cameras tested, the pixel-to-pixel variations in the flatfield map were very little (Supplementary Fig. 5). Note that our approach does not correct for possible nonlinearities in the camera (Supplementary Fig. 6), which would require more extensive characterization and correction routines[19].

**Dark current correction in 3D super-resolution microscopy.** Precise knowledge of the relevant (s)CMOS characteristics encoded in the offset, noise and photon response maps is crucial for accurate fitting in single molecule localization microscopy[1,5] (SMLM). This holds especially true in the vicinity of hot pixels, which show increased offset and noise that strongly depend on the exposure time (Supplementary Fig. 7). (s)CMOS-specific SMLM fitters[1,6] which are based on maximum likelihood estimation (MLE)[20] can achieve the theoretically achievable precision as given by the square root of the Cramer-Rao lower bound (CRLB). The camera maps generated by our approach (Fig. 1) result in formally correct consideration of both dark current and thermal noise in (s)CMOS specific fitting (Supplementary Note 1) and we integrated the workflow into our SMLM software SMAP[21].

To visualize the effect of (s)CMOS characteristics on SMLM, we simulated experiments of astigmatism-based[22] 3D SMLM using measured maps of a latest generation, cooled scientific-

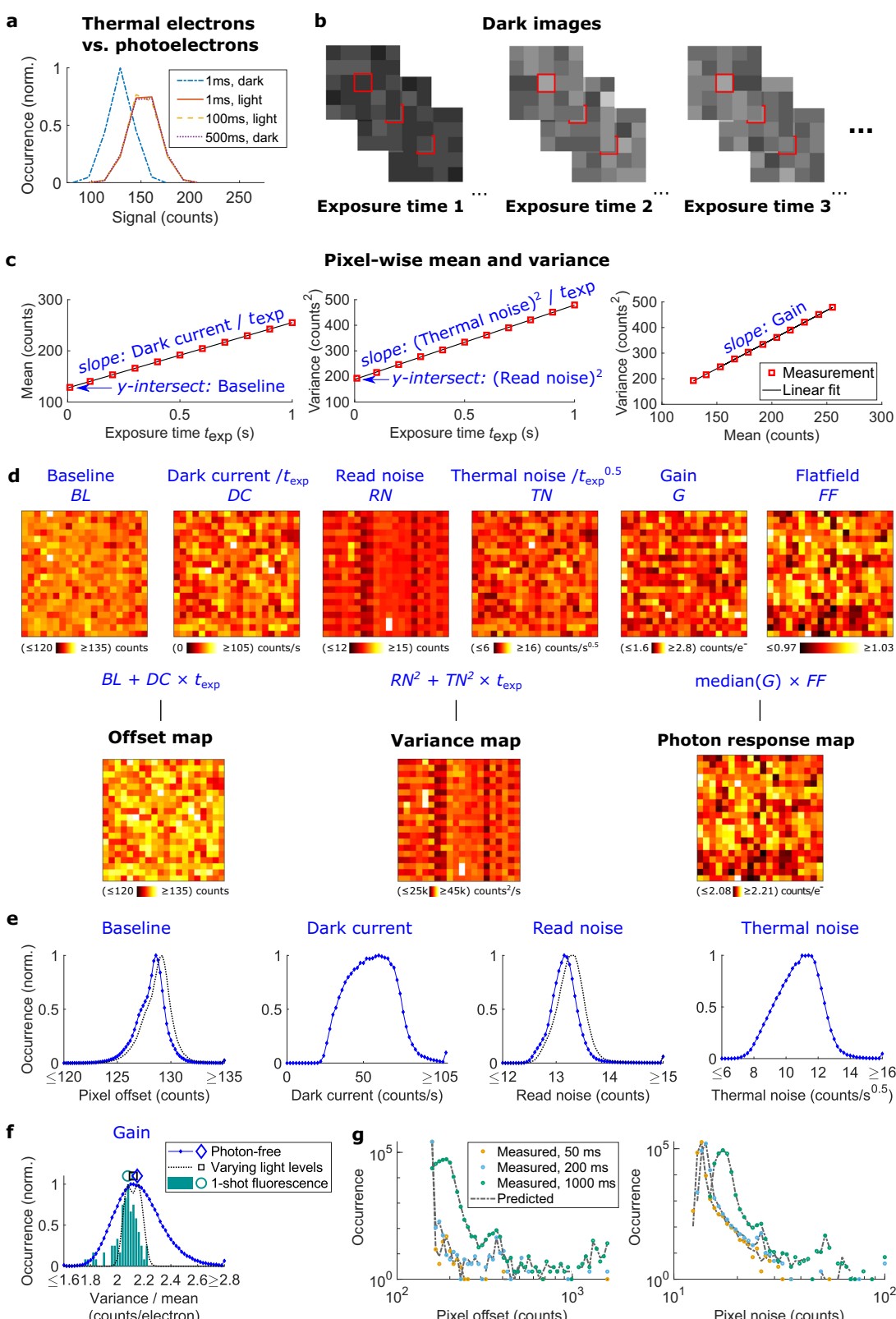

grade CMOS camera (Fig. 2a, b) and typical fluorophore parameters for traditional DNA point accumulation in nanoscale topography (DNA-PAINT)[23] (*i.e.* long exposure time and high photon numbers), (direct) stochastic optical reconstruction microscopy (STORM)[24] (i.e. short exposure time and medium photon numbers), and photoactivated localization microscopy (PALM)[25] (i.e. short exposure time and low photon numbers) (Methods). When (s)CMOS specific fitting is not applied, regions close to pixels of high dark current show a high bias in the 3D localization coordinates (Fig. 2c, e), even for DNA-PAINT for which sCMOS specific SMLM fitting is often neglected[26]. Application of sCMOS specific fitting largely removes the bias (Fig. 2d, f, g), and restores the theoretically achievable root mean square error (Fig. 2h) for all SMLM modalities.

**Fig. 1 Automated camera characterization via thermally generated signal. a** Signal statistics of a single pixel at different conditions showing mostly read noise (1 ms exposure time and no light), read noise and thermally generated signal (500 ms and no light), read noise and photon generated signal (1 ms and light) as well as read noise, thermally generated and photon generated signal (100 ms and less light). **b–d** Workflow of camera characterization. **b** A series of dark images is automatically recorded at different exposure times. For each pixel and exposure time the mean and variance of the signal is calculated. **c** Result for one pixel of an uncooled CMOS camera. Dark current and thermal noise squared are proportional to the exposure time, so the temporal dependence can be determined from the slope of linear fits. The y-intersects of the fits reveal the baseline as well as the read noise squared, free from thermal effects. Since thermally generated signal follows Poisson statistics, the variance is proportional to the mean, with the proportionality factor corresponding to the pixel gain. **d** Baseline, dark current, read noise, thermal noise and gain maps are calculated pixel-wise as in (**c**). Optionally, we acquire a single bright image for flat-field correction. From these maps, we calculate the exposure-time dependent offset, variance and photon response maps. These maps can be used as input for existing camera correction algorithms for images recorded at arbitrary exposure times. **e** Histograms of pixel values obtained by photon-free characterization and traditional characterization of using varying light levels. The traditional characterization overestimates baseline and read noise by the thermal effects for the corresponding exposure time. **f** Distribution of the gain determined via different approaches (pixelwise histogram for the photon-free and varying light levels approaches, histogram of outcomes from multiple determinations of the mean gain from the 1-shot approach). Symbols above the curves indicate the medians. **g** Comparison of pixel offset and noise distributions from dark frames at different single frame exposure times either predicted using the calculations shown in (**d**), or directly determined from pixel-wise means and standard deviations.

To validate our simulation results with experimental data, we performed 3D DNA-PAINT[23] using the same cooled scientific-grade CMOS camera. One might expect the bright fluorescence signal to be significantly higher than thermally generated signal. However, emitter dwell-times are in the 1 s regime, so dark current can play a pronounced role. Our experiments (Fig. 2i, j) confirm the simulation results that the proximity of uncorrected high dark-current pixels leads to shifts in the localized coordinates. Although high dark-current pixels are relatively sparse on cooled, scientific-grade cameras, this bias locally misplaces structures in 3D (Fig. 2k, l), easily exceeding nanometer localization precisions[26]. As expected from the simulations, the resulting distortion (Fig. 2k-n, Supplementary Video 1) is highly spatially dependent and changes its direction over only a few hundred nanometers. Consequently, even cooled sCMOS cameras should be characterized carefully and corrected for thermal effects for unbiased SMLM reconstructions. Besides DNA-PAINT, such characteristics (*i.e.* long exposure times and high photon numbers) are also relevant for STORM under resolution-optimized conditions[27].

**Correction of uncooled CMOS cameras.** We next investigated if our approach can render uncooled, economic industry-grade CMOS cameras (Supplementary Fig. 2) usable for high-quality 3D SMLM. Compared to sCMOS cameras, industry-grade CMOS cameras show higher dark current, higher noise and generally less uniform pixel properties[9] (Fig. 3a, b, all characteristics shown in Fig. 1e–g). In simulations, these lead to an even larger bias in the localizations. Especially for PALM, local bias exceeded 50 nm laterally and 150 nm axially (Fig. 3c, e). Again, consideration of pixel-dependent effects removes the bias (Fig. 3d, f, g) and restores the theoretically achievable root mean square error (Fig. 3h) for PALM, STORM and DNA-PAINT. Notably, applying general CMOS corrections but without explicit consideration of the exposure time retains bias (Supplementary Fig. 8).

The potential of an uncooled, industry-grade CMOS camera becomes visible when we used it for 3D PALM[25], *i.e.* SMLM with photoconvertible fluorescent proteins. PALM has lower signal levels as compared to STORM and DNA-PAINT, which is particularly challenging in presence of camera noise and for 3D SMLM. Following our photon-free camera characterization and applying the camera maps in CMOS-specific fitting, we could well resolve the 3D structure of clathrin coated pits by the same low-cost camera (Fig. 3i-j). 3D STORM[24] (Fig. 3k) on the nucleoporin Nup107 (ref. [28]) clearly resolved individual corners of the nuclear pore complex (Fig. 3l) in the lateral projection and parallel lines in the axial projection (Fig. 3m) stemming from the nucleo- and cytoplasmic rings. This indicates a resolution better than 57 nm in the axial direction[29], achieved with this uncooled, but properly characterized industry-grade camera. Thus, we could show that with our approach, low-cost cameras exhibit only slightly reduced performance compared to sCMOS cameras, an important development in light of their recent popularity for SMLM[8–10,12,14,15].

**Diffraction-limited image restoration.** Our CMOS characterization pipeline can also help in diffraction limited image restoration. Liu et al. have presented a noise correction algorithm (NCS)[3] and Mandracchia et al. have presented an algorithm for automatic correction of sCMOS-related noise (ACsN)[4]. The aim of both approaches is to mitigate the effect of the (s)CMOS detector on wide-field images while preserving the characteristics of the fluorescence signal. Hence, both approaches rely on a proper camera characterization. NCS uses the three camera maps of offset, gain and noise. ACsN uses the two camera maps of offset and gain. We used an accordingly characterized, uncooled CMOS camera to image AP-2 in live U373 cells via total internal reflection fluorescence microscopy (TIRF)[30] (Supplementary Video 2). The raw data shows numerous pixels of strongly increased offset and noise (Fig. 3n). Both NCS (Fig. 3o) and ACsN (Fig. 3p) removed noise and bias of such bad pixels (Fig. 3q) and considerably increased the image quality.

**Software implementation.** Besides the conventional (s)CMOS corrections for pixel specific noise, gain and offset, our results indicate the benefit of characterizing and correcting for the effects of dark current and associated thermal noise in high- and super-resolution microscopy, both for uncooled and cooled cameras. To make our approach easily accessible for the imaging community, we implemented the *automated camera characterization* via *electron noise tool* (ACCeNT) in the popular open-source software Micro-Manager[16] and ImageJ/Fiji[17] (Supplementary Note 2). All relevant camera properties, including thermal effects, can be determined without user intervention and there is no need for additional hardware implementations. Thus, existing algorithms that demand proper camera characterizations, like the ones we used in this work, become applicable to the broad audience.

## Methods

**Camera calibration.** For the photon-free calibration, all light to the camera chip was blocked by screwing a lid to the camera mount. Before starting the measurement, the camera was pre-run to give the detector time to thermally equilibrate either to the targeted cooling temperature (−10 °C as this was the manufacturer's calibration setpoint) or warming up to the operating temperature

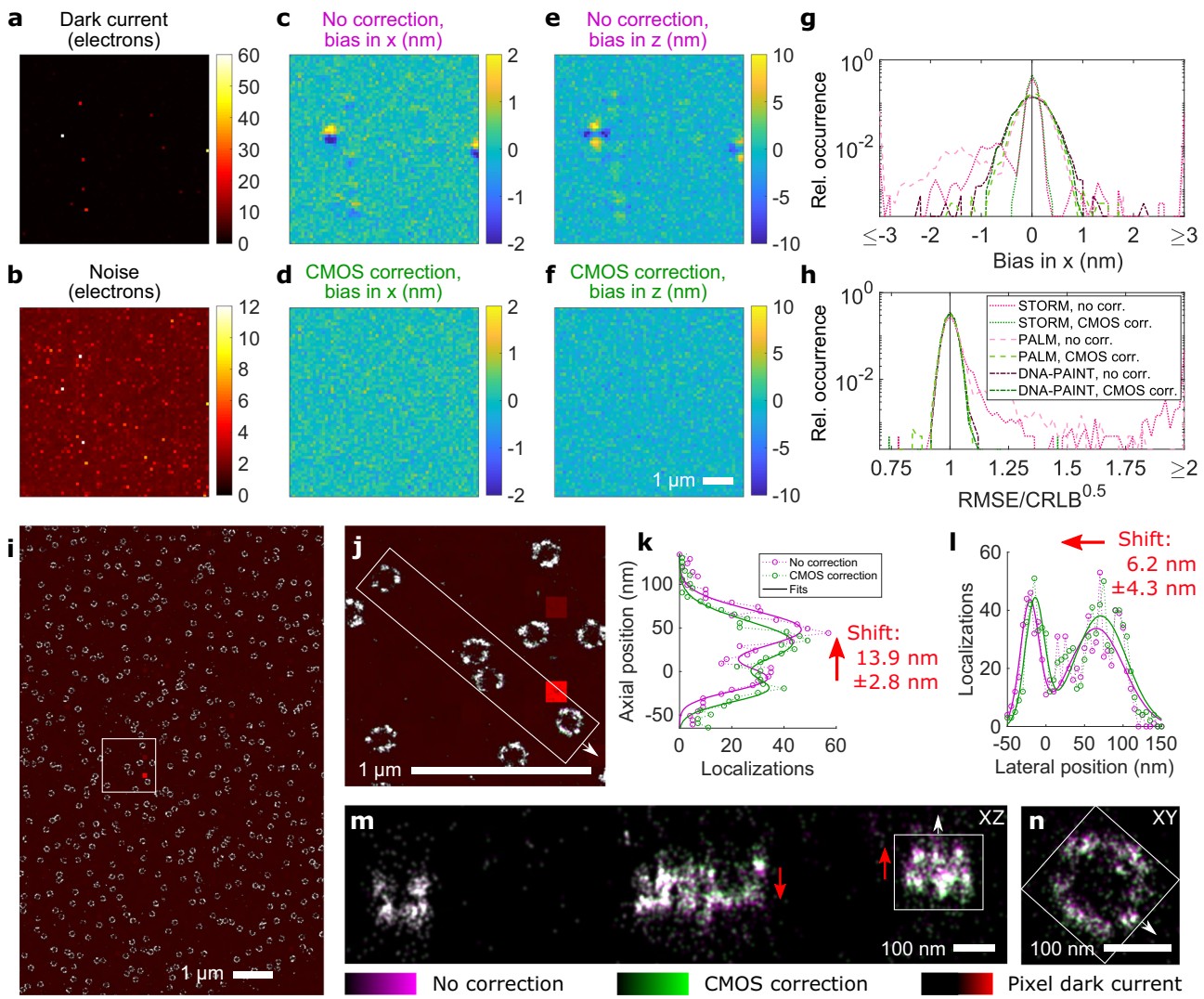

**Fig. 2 Camera calibration for thermal effects circumvents systematic fitting errors in SMLM.** Maps of dark current (**a**) and noise (**b**) at 500 ms single frame exposure time for a scientific-grade CMOS (sCMOS) camera cooled to the manufacturer's calibration setpoint of −10 °C. Simulations of 3D DNA-PAINT via astigmatism-based PSF shaping using this camera reveal a particular pattern in the localization bias close to pixels of high dark current, both laterally (**c**) and axially (**e**) when not applying (s)CMOS-specific fitting that corrects for pixel-wise effects including thermal effects. Explicit application of (s)CMOS specific fitting largely removes the bias for DNA-PAINT (**d**, **f**) as well as STORM and PALM (**g**) and restores the theoretically achievable root mean square error in the localizations (**h**). **i**, Experimental 3D DNA-PAINT data of the nucleoporin Nup96 in U2OS cells using the same cooled sCMOS camera. The image is rendered as an overlay of the pixel dark current map (red) and the SMLM reconstruction with no camera correction (magenta) and with CMOS correction (green). **j**, Zoom into boxed region in **i**. **k**, **l**, The structure of a nuclear pore complex (indicated by the boxes in (**m**), (**n**)) becomes shifted in the vicinity of a pixel of high dark current, both in axial (**k**) and lateral (**l**) direction when neglecting individual pixel characteristics including thermal effects in the fitting pipeline. **m** Axial view of the region indicated in (**j**), also shown in Supplementary Video 1. **n** Lateral close up of the nuclear pore complex indicated in (**m**). As expected from the simulation, the shift features a high spatial dependence (**m**, **n**), which even changes its sign (indicated by the red arrows in **n**).

in case of uncooled cameras. We acquired around 8,000 to 20,000 sets of typically 5 to 10 different exposure times. To maintain a constant average detector temperature, recording was performed in a nested manner, i.e. we changed the exposure time after each camera frame and then repeated acquisition of all exposure times.

Initially, we used custom-written scripts in Micro-Manager, Fiji and MATLAB (Mathworks) for data acquisition and analysis, but later implemented the entire workflow into independent Micro-Manager and Fiji ACCeNT plugins (see next paragraph). After recording of raw data as described in the former paragraph, Fiji was used to process the TIFF stacks. For each exposure time and pixel, the mean value and standard deviation were calculated and saved as TIFF files: One after another, the stack corresponding to one exposure time was imported to Fiji and the "z-project" function was called with projection type "Average Intensity" for the mean value or projection type "Standard Deviation" for the standard deviation. The resulting TIFF files were imported into MATLAB for further processing. For each pixel linear functions were fitted using the polyfit function to (i) the mean value as a function of the exposure time to

determine the baseline from the y-intersect and the dark current per time from the slope, (ii) the variance (i.e. the standard deviation squared) as a function of the exposure time to determine the read noise squared from the y-intersect and the thermal noise squared per time from the slope, and (iii) the variance as a function of the mean value to determine the gain (i.e. the conversion factor from electrons to ADU counts) from the slope. For each pixel, the exposure time dependent offset was calculated as the baseline plus the dark current per time multiplied by the single frame exposure time. For each pixel, the exposure time dependent noise squared was calculated as the read noise squared plus the thermal noise squared per time multiplied by the single frame exposure time. For each pixel, the photon response was calculated as the median of the gain map for all pixels multiplied by the pixel-wise value of the flatfield map. To find the flatfield map, we exposed the camera to a homogeneous illumination via ambient light and applied the algorithm presented by Lin et al.[5].

We implemented the photon-free calibration workflow including the automated nested data acquisition, fitting of individual pixel properties and calculation of exposure time dependent camera maps as the ACCeNT plugin for Micro-Manager 2.

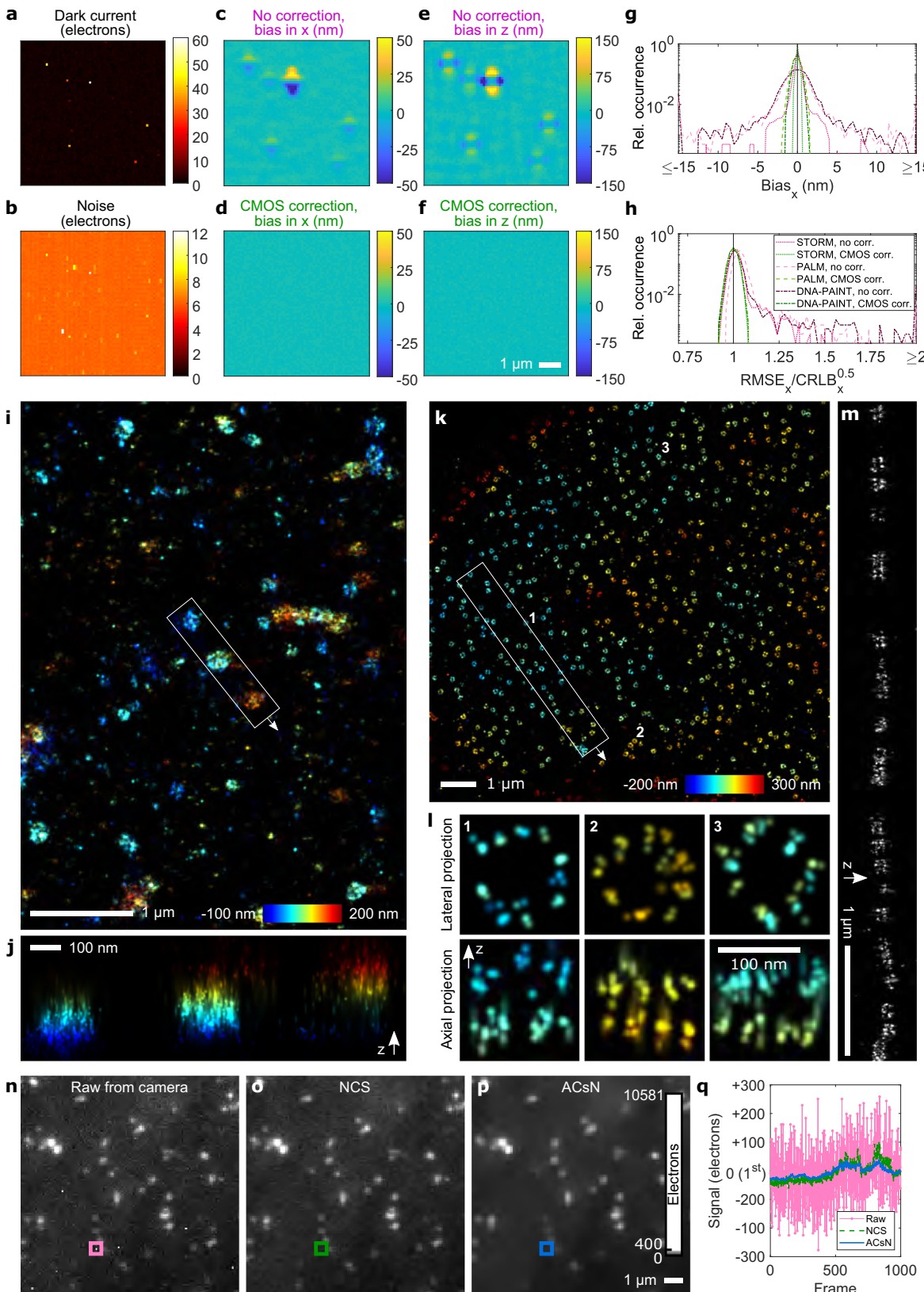

Additionally, we implemented the fitting of individual pixel properties and calculation of exposure time dependent camera maps as an ACCeNT plugin for Fiji. The latter is intended to be used for processing of data acquired with software different from Micro-Manager 2, e.g. if the microscope is run using Micro-Manager 1.4 or the manufacturer's software. For Micro-Manager 1.4 users, we provide a script for the automated nested data acquisition. We checked the consistency of all implementations against each other.

**Microscope.** All data was acquired on a custom-built microscope as described in the following. Laser light was emitted from the single mode fiber output of a laser box (iChrome MLE, Toptica) (640 nm for excitation of Alexa Fluor 647 and Atto 655 and TetraSpeck beads, 561 nm for excitation of mEos and TetraSpeck beads, 488 nm for excitation of GFP and 405 nm for active photoswitching in STORM and PALM experiments) and collimated using an achromatic lens (either f = 50 mm for DNA-PAINT imaging using the industry-grade CMOS camera or f = 30 mm for

**Fig. 3 Camera calibration increases performance of an uncooled, industry-grade CMOS camera for SMLM and diffraction-limited fluorescence imaging.** Maps of dark current (**a**) and noise (**b**) at 50 ms single frame exposure time for an uncooled, industry-grade CMOS camera (characteristics shown in Fig. 1d, e). Simulations of astigmatism-based 3D PALM without explicit consideration of pixel-wise effects show a similar pattern in the localization bias, but of greater amplitude as compared to the cooled sCMOS camera (compare Fig. 2), both laterally (**c**) and axially (**e**). Explicit application of CMOS-specific fitting largely removes the bias for PALM (**d**, **f**) as well as STORM and DNA-PAINT (**g**) and restores the theoretically achievable root mean square error in the localizations (**h**). **i**, Experimental 3D PALM data of clathrin tagged with mEOS3.2 in a U2OS cell using the same camera and applying CMOS-specific fitting including thermal effects. **j**, Axial view of region indicated in **i**. **k** Experimental 3D STORM data of Nup107-SNAP, labeled with AF 647, in a U2OS cell using the same camera and applying CMOS-specific fitting including thermal effects. **l** Gallery showing lateral and axial views on individual nuclear pore complexes indicated in **k**. **m** The axial view on the region indicated in (**k**) shows two parallel lines from the nucleo- and cytoplasmic rings 57 nm apart. **n** First frame of experimental raw data from time-lapse TIRF imaging of AP-2 tagged with eGFP in live U373 cells recorded at 1000 ms single frame exposure time. **o** NCS corrected and (**p**) ACsN corrected frame. The entire time-lapse for (**n–p**) is shown in Supplementary Video 2. **q** The noise of a pixel of high dark current is strongly reduced via both approaches after appropriate characterization of the camera including thermal effects. The signal has been offset-corrected by the pixel value of the first frame from the time-lapse.

live-cell experiments and all STORM and PALM experiments, all lenses from Thorlabs) or emitted from a fiber laser (F-04306-107, MPB Communications) (642 nm for excitation of Atto 655), filtered through an AOTF (AA Opto Electronic) and expanded using telescope of achromatic lenses (f = 50 mm and f = 100 mm, both Thorlabs). The collimated laser light was focused (f = 150 nm, Thorlabs) to the back focal plane of a TIRF objective lens (either 100x, NA 1.35 silicone oil, Olympus for DNA-PAINT using a cooled sCMOS camera, or 60x, NA 1.49 oil, Olympus for all other experiments). Imaging the fiber output in 4f-configuration and mounting it to a linear stage (SLC2445me-4, Smaract) enabled image acquisitions in epi, HILO and TIRF illumination. The resulting projected pixel widths were 98 nm for an uncooled, industry-grade CMOS camera (µeye UI-3060CP-M-GL R, IDS), 58 nm for a different uncooled, industry-grade CMOS camera (Chameleon3 CM3-U3-50S5, FLIR), and 117 nm for a cooled, scientific-grade CMOS (sCMOS) camera (Edge 4.2 bi, PCO).

Fluorescence emission was separated from the laser excitation via a dichroic beamsplitter (zt405/488/561/640rpc, Chroma), further filtered (either bandpass 697/58, Semrock for DNA-PAINT using a cooled sCMOS camera; bandpass 700/100, Chroma plus notch filter 400-410/488/561/631-640, Semrock, for DNA-PANT using an uncooled, industry-grade CMOS camera; bandpass 676/37, Semrock for STORM; longpass 568, Semrock, plus bandpass 600/60, Chroma for PALM imaging; or 525/50, Semrock for live-cell GFP imaging) and focused onto the camera by a tube lens (either f = 100 mm, Thorlabs for DNA-PAINT using the cooled sCMOS camera; or f = 180 mm, Olympus for all other experiments). For 3D SMLM experiments via astigmatism-based PSF shaping, a cylindrical lens was placed before the camera (f = 2000 mm, CVI Laser Optics). An additional short pass filter (FESH750, Thorlabs) was used before the camera to block light from a focus lock laser. The focus lock laser (785 nm, Toptica) was coupled into the excitation beam path using an additional dichroic mirror, reflected off the coverslip-buffer interface of the sample, and its position was detected using a four-quadrant photodiode. The photodiode output was used to maintain the z-position of the objective lens constant with respect to the sample for active z-drift compensation.

The microscope hardware and data acquisition was handled via Micro-Manager 1.4.22 using custom-written software[31]. When imaging using an industry-grade CMOS camera, the excitation laser was run constantly. When imaging using the cooled, scientific-grade sCMOS camera, the excitation laser was triggered on during the common exposure of all lines of the camera. In all cases, the UV laser for active photoswitching in PALM and STORM experiments was triggered at the camera frame rate, but the pulse length was dynamically adjusted to aim for a constant number of active emitters per frame.

### Sample preparation

*U2OS cells NUP107-SNAP for STORM.* U2OS Nup107-SNAP samples stained with Alexa Fluor 647 for STORM imaging were prepared as previously described[29]. U2OS NUP107-SNAP-tag cells (catalog no. 300294, CLS Cell Line Service, Eppelheim, Germany) were seeded onto clean 24 mm round glass coverslips and grown in phenol-red free Dulbecco's Modified Eagle Medium growth medium (DMEM, Gibco no. 11880-02; 1x MEM NEAA, Gibco no. 11140-035; 1x Gluta-MAX, Gibco no. 35050-038; 10% [v/v] fetal bovine serum, Gibco no. 10270-106). For nuclear pore staining, the coverslips were rinsed twice with PBS and prefixed with 2.4% [w/v] FA in PBS for 30 s. Cells were permeabilized with 0.4% [v/v] Triton X-100 in PBS for 3 minutes and afterwards fixed with 2.4% [w/v] FA in PBS for 30 minutes. Subsequently, the fixation reaction was quenched by incubation in 100 mM NH4Cl in PBS for 5 minutes. After washing twice with PBS, the samples were blocked with Image-iT FX Signal Enhancer (ThermoFisher Scientific, Waltham, MA, USA) for 30 min. The coverslips were incubated in staining solution (1 µM benzylguanine Alexa Fluor 647 (S9136S, NEB, Ipswich, MA, USA); 1 mM DTT; 1% [w/v] BSA; in PBS) for 50 minutes in the dark. After rinsing three times with PBS and washing three times with PBS for 5 min, the sample was mounted for imaging.

For STORM imaging, coverslips were mounted in 500 µL blinking buffer (50 mM) Tris pH 8, 10 mM NaCl, 10% [w/v] D-glucose, 35 mM 2-mercaptoethylamine (MEA), 500 µg/mL GLOX, 40 µg/mL catalase.

*U2OS cells clathrin mEOS3.2 for PALM.* U2OS cells (U2OS NUP96-SNAP-tag cell line) were seeded onto clean 24 mm round glass coverslips and grown in phenol-red free Dulbecco's Modified Eagle Medium growth medium (DMEM, Gibco no. 11880-02; 1x MEM NEAA, Gibco no. 11140-035; 1x GlutaMAX, Gibco no. 35050-038; 10% [v/v] fetal bovine serum, Gibco no. 10270-106) (cell culture and seeding conditions described in ref. [28]). Transient transfection with a clathrin-mEOS3.2 construct (Addgene 57452) was achieved using Lipofectamine™ 2000 reagent (Life Technologies) according to the manufacturer's recommendations: DNA (1 µg) was mixed with OptiMEM I (50 µL), and Lipofectamin (3 µL) was mixed with Opti-MEM I (50 µL). Both solutions were incubated for 3 min at room temperature, mixed together and incubated for additional 10 min at room temperature. After exchanging the culture medium with prewarmed OptiMEM I, the DNA-Lipofectamin solution (100 uL) was added dropwise to the seeded cells. After approximately 24 h incubation (at 5% CO₂, 37 °C), the medium was exchanged with fresh growth medium. After additional incubation for approximately 24 hours, cells were fixed for 20 min in 3% [w/v] paraformaldehyde in cytoskeleton buffer (CB; 10 mM MES pH 6.1, 150 mM NaCL, 5 mM EGTA, 5 mM D-glucose, 5 mM MgCL₂, as described in ref. [32]) at room temperature. The fixation process was stopped by incubation for 7 min in 0.1% [w/v] NaBH₄ at room temperature. The sample was washed 3 times for 5 min in PBS.

*U373 cells AP2-eGFP for live-cell TIRF.* U373 cells stably expressing AP2-eGFP (generously provided by the Boulant lab, German Cancer Research Center (DKFZ), Heidelberg) were cultured in high glucose growth medium (DMEM, Gibco no. 11880-02; 1x MEM NEAA, Gibco no. 11140-035; 1x GlutaMAX, Gibco no. 35050-038; 10% [v/v] fetal bovine serum, Gibco no. 10270-106; 20% [w/v] glucose; 1x ZellShield, Minerva Labs) at 37 °C and 5% CO₂. Passaging was done every 2-3 days to maintain the cells at approximately 50% confluency.

*U2OS cells immunostained for DNA-PAINT.* U2OS cell immunostained for microtubules for DNA-PAINT imaging were prepared as previously described[6]. In brief, U-2 OS wild type cells were prefixed for 2 min with 0.3% (v/v) glutaraldehyde in cytoskeleton buffer (CB; 10 mM MES, pH 6.1, 150 mM NaCl, 5 mM EGTA, 5 mM d-glucose, 5 mM MgCl₂) + 0.25% (v/v) Triton X-100 and fixed with 2% (v/v) glutaraldehyde in CB for 10 min. Fluorescent background was reduced by incubation with 0.1% (w/v) NaBH₄ in PBS for 7 min. After samples had been washed three times with PBS, microtubules were labeled with anti-β-tubulin antibody (T5293; Sigma-Aldrich), diluted 1:300 in PBS with 2% (w/v) BSA, for 2 h. After being washed three times with PBS, samples were incubated with a DNA-labeled anti-mouse secondary antibody overnight (docking strand sequence: 5′-TTATACATCTA-3′) and imaged after 5 washes with PBS using 50 pM of complementary Atto-655-labeled DNA imager strand (5′-CTAGATGTAT-3′-Atto655) in PAINT buffer (PBS, 500 mM NaCl, 40 mM Tris, pH 8.0).

*U2OS cells Nup96-eGFP for DNA-PAINT.* Cells were seeded as previously described on high-precision 24 mm round glass coverslips[33]. In short, coverslips (No. 1.5H, catalog no. 117640, Marienfeld) were cleaned in a methanol:hydrochloric acid (50:50) mixture overnight before washing them repeatedly with ddH₂O and drying them in a laminar flow hood. Before usage, clean coverslips were additionally irradiated with UV for 30 min.

U2OS Nup96-mEGFP cells were seeded onto the coverslips in such a density, that they reach a confluency of 50 to 70% on the day of fixation (typically 2 days after seeding). During this time, cells were grown in an incubation chamber providing 37 °C and 5% CO₂ in growth medium (DMEM (catalog no. 11880-02, Gibco)) containing 1 × MEM NEAA (catalog no. 11140-035, Gibco), 1× GlutaMAX (catalog no. 35050-038, Gibco) and 10% (v/v) fetal bovine serum (catalog no.

10270-106, Gibco). Finally, shortly before fixation, coverslips were rinsed twice with warm PBS.

Coverslips containing U2OS Nup96-mEGFP cells (catalog no. 300174, CLS Cell Line Service, Eppelheim, Germany) were first prefixed in 2.4% w/v formaldehyde (FA) in PBS for 40 s before samples were incubated in 0.1% v/v Triton X-100 in PBS for 3 min. After washing samples twice for 5 min in PBS, fixation was completed in 2.4% w/v FA in PBS for 20 min. The sample was subsequently washed twice in PBS for 5 min each before remaining FA was quenched in 100 mM NH₄Cl in PBS for 5 min and then washed twice in PBS for 5 min. Permeabilization was carried out in 0.2% v/v Triton X-100 in PBS and remaining permeabilization solution was washed away twice in PBS for 5 min each. Samples were blocked in 2% w/v BSA in PBS for 1 h, before coverslips were placed upside down onto a drop of primary antibody staining mix (rabbit anti-GFP, catalog no. 598, MBL International, diluted 1:250 in PBS containing 2% w/v BSA) overnight at 4 °C. Weakly and unbound primary antibodies were washed off thrice in PBS for 5 min each. Similarly, binding of anti-rabbit secondary i1 (docking strand sequence: 5'-TTATACATCTA-3') DNA-PAINT antibodies (homemade, kind gift of Ingmar Schoen, Royal College of Surgeons in Ireland) was achieved by placing the samples upside down onto a 1:100 dilution of the antibodies in PBS containing 2% w/v BSA for 1 h at RT. After washing thrice in PBS for 5 min each, a post-fixation was carried out in 2.4% w/v FA for 30 min. Samples were washed twice for 5 min in PBS and finally placed into a custom-made sample holder.

*Fluorescent bead samples.* 100 nm sized TetraSpeck beads (Thermo Fisher) were diluted 1:40 in 100 mM MgCl2 in H2O and incubated for 3 min on coverslips. Before imaging and PSF calibration via z-stacks, the bead solution was replaced by H₂O.

*Data acquisition.* PALM imaging was performed using an uncooled, industry-grade CMOS camera (μeye UI-3060CP-M-GL R, IDS). Fixed U2OS cells were imaged in buffer containing 95 % D2O and 50 mM Tris/HCl pH9. Raw data was acquired in HILO illumination at 561 nm and laser output powers of 20 mW to 50 mW. The single frame exposure time was set to 50 ms.

STORM imaging was performed using either an uncooled, industry-grade CMOS camera (μeye UI-3060CP-M-GL R, IDS) or a different uncooled, industry-grade CMOS camera (Chameleon3 CM3-U3-50S5, FLIR). Fixed U2OS cells were imaged in blinking buffer containing 50 mM Tris/HCl pH8, 10 mM NaCl, 10% (w/v) D-glucose, 500 μg/ml glucose oxidase, 40 μg/ml catalase, 143 mM BME and 2 mM COT. Raw data was acquired in HILO illumination at 640 nm and at a laser output power 70 mW. The single frame exposure time was set to 50 ms.

DNA-PAINT imaging of tubulin in U2OS cells was performed using an uncooled, industry-grade CMOS camera (μeye UI-3060CP-M-GL R, IDS). Fixed U2OS cells were imaged in buffer containing 500 mM NaCl, 1x PBS, 40 mM Tris/HCl pH8 and imager strands (I1, Ultivue, 5'-CTAGATGTAT-3'-Atto655) at a concentration of about 500 pM. Raw data was aquired in HILO illumination at 640 nm and a laser output power of 70 mW. The single frame exposure time was set to 500 ms.

DNA-PAINT imaging of Nup96 in U2OS cells was performed using a cooled, scientific-grade CMOS (sCMOS) camera (Edge 4.2bi, PCO). Fixed U2OS cells were imaged in buffer containing 500 mM NaCl, 40 mM Tris/HCl pH8 and imager strands (I1 650, i.e. Atto 655, Ultivue) at a concentration of about 500 pM. Raw data was acquired in HILO illumination at 642 nm and a laser output power of 4.5 mW. The single frame exposure time was set to 500 ms.

Diffraction limited TIRF imaging of AP-2 in U373 cells was performed using an uncooled, industry-grade CMOS camera (μeye UI-3060CP-M-GL R, IDS). Live U373 cells were imaged at room temperature in growth medium. Raw data was acquired in shallow TIRF illumination at 488 nm and a laser output power of 0.1 mW. The single frame exposure time was set to 1000 ms.

*Image data analysis.* SMLM data was fitted and analyzed as previously described[28] using our custom-written, open-source superresolution microscopy analysis platform SMAP[21] in MATLAB. The software is available at github.com/jries/SMAP. In case of (s)CMOS specific fitting, the predetermined camera maps were applied for the exposure time of the respective experiment.

3D SMLM data (STORM, PALM, DNA-PAINT) was fitted with an experimentally derived PSF model measured via z-stacks of 100 nm sized fluorescent beads as previously described[6]. For STORM data, the localizations were filtered for a lateral localization precision better than 12.7 nm, a relative log-likelihood value better than −2.9, and the first 600 frames were filtered out. For PALM data, the localizations were not further filtered. For DNA-PAINT data, the localizations were filtered for a localization precision from 0 to 12 nm and a z-coordinate of 200 nm to 100 nm. 2D DNA-PAINT data was fitted with a Gaussian PSF model and the localizations were filtered for a localization precision better than 30 nm and a PSF width of 100 to 175 nm. Diffraction-limited TIRF images were processed using the NCS software and ACsN software, respectively, as provided by the authors. As input, we use the camera maps determined via the photon-free approach described in this work, encoding the pixel-wise properties for gain, offset, and noise in case of NCS and gain and offset in case of ACsN. We parameterized the NCS MATLAB "single pixel with normalization"-algorithm by an alpha weight factor of 10, a pixel size of 0.0977 μm, an emission wavelength of

0.525 μm, a numerical aperture of 1.49, and 25 iterations. We parameterized the ACsN MATLAB app with a numerical aperture of 1.49, an emission wavelength of 525 nm and a pixel size of 108 nm, turned the video filter off and the parallel CPU option on.

*SMLM simulation.* Raw 3D SMLM data were simulated in MATLAB using an experimentally derived PSF model for the microscope described above, experimentally derived camera characteristics via the photon-free approach described in this work, and photon counts parameterized by DNA-PAINT, STORM and PALM experiments described above. Camera data was simulated using a projected camera pixel width of 98 nm and the emitters were placed on the center of each camera pixel. Each emitter position was simulated for 1,000 times with the distribution of photon counts drawn from the experimentally derived distribution of the photon counts per emitter and per frame. Poisson noise was added to the photon distribution over the experimental PSF and the fluorescence signal was converted to ADU counts. The camera baseline was added, the dark current was added corresponding to the respective single frame exposure time (50 ms for PALM and STORM, 500 ms for DNA-PAINT), read noise and thermal noise was added corresponding to the respective single frame exposure time. The synthetic raw 3D SMLM data was then fitted either using a (s)CMOS-specific fitter with explicit consideration of pixel-to-pixel variations of the camera properties including dark current and thermal noise, or neglecting pixel-to-pixel variations and using the average values of the camera properties instead. The bias for each emitter position was determined as the deviation of the mean fitted coordinate from the ground truth.

For the expected root mean square error (RMSE), we followed the same approach as described above, but did not draw the photon counts from a distribution. Instead, we simulated all emitters with the same photon counts using the mean photon counts from the distribution (*i.e.* 3,420 photons for PALM, 9,000 photons for STORM, 35,100 photons for DNA-PAINT in case of the scientific-grade CMOS camera, and 1,900 photons for PALM, 5,000 photons for STORM, 19,500 photons for DNA-PAINT in case of the industry-grade CMOS camera). The theoretically achievable precision was calculated via the square root of the Cramér-Rao lower bound (CRLB)[20] according to the particular PSF shape[6].

**Reporting Summary.** Further information on research design is available in the Nature Research Reporting Summary linked to this article.

## Data availability
Example data for testing the ACCeNT software implementation can be downloaded from https://rieslab.de/#accent. All other data are available upon request from the corresponding author.

## Code availability
The software for the data acquisition and analysis used in this paper is available at https://github.com/ries-lab/Accent/releases and https://github.com/jries/SMAP[34].

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

## Acknowledgements

We thank Ingmar Schoen for providing secondary antibodies for the DNA-PAINT experiments and Steve Boulant for providing the U373 cell line. We thank Alejandro Colchero for assistance in setting up the microscope, Philipp Hoess and Jervis The-vathasan for additional sample preparations, Amir Rahmani for assistance in data acquisition and Sheng Liu and Anders Engdahl for testing of the software. This work was supported by the European Research Council (grant no. ERC CoG-724489 to JR), the National Institutes of Health Common Fund 4D Nucleome Program (grant no. U01 EB021223 to JR), the Human Frontier Science Program (RGY0065/2017 to JR), the Engelhorn Foundation (Postdoctoral Fellowship to RD) and the European Molecular Biology Laboratory.

## Author contributions

R.D. conceived the project, recorded the data, analyzed the data and prepared the figures. J.R. supervised the project and developed the SMAP software for SMLM reconstruction and analysis. J.D. programmed the ACCeNT-specific plugins and scripts for Micro-Manager and Fiji. Y.L. performed the simulations. T.D. recorded camera characterization data. A.T. prepared the live-cell and PALM samples. M.K. prepared the DNA-PAINT samples. UM prepared the STORM samples. R.D., J.R., J.D., Y.L., and T.D. prepared the manuscript with input from all authors. All authors reviewed the manuscript.

## Funding

## Competing interests

The authors declare no competing interests.
