## [Peer Review File · Nature Communications]

Photon-free (s)CMOS camera characterization for artifact reduction in high- and super-resolution microscopyREVIEWER COMMENTS

Reviewer #1 (Remarks to the Author):

This manuscript concisely describes a method of pixel-wise calibration for (s)CMOS cameras, which are becoming ubiquitous in fluorescence microscopy including super-resolution applications. The method uses pixel-wise thermal noise and different exposure times to find gain, offset, readnoise and thermal noise. Readnoise and thermal noise can be combined to give a total camera noise, while thermal noise average and the offset can be combined to give a pixel-wise offset. The paper is very complete, even showing the capability with noise CMOS camera. The questions I would have on performance have been addressed in the manuscript.

One small detail I (as a potential user) would like to know is for how long does the calibration remain valid? Although it does seem simple enough to do daily, or just before every long SR data acquisition.

I will likely implement this method in our setup.

Reviewer #2 (Remarks to the Author):

In their manuscript entitled "Photon-free (s)CMOS camera characterization for artifact reduction in high- and super-resolution microscopy", Diekmann et al. develop a new (and potentially smart) method for calibration of CMOS cameras of both scientific and industry grade. The novelty of the approach is the fact that thermally generated dark current rather than photonic input is used as the basis for calibration. Consequently, only dark images obtained with the shutter closed at various exposure times are needed, which simplifies data acquisition. It is well-known that CMOS cameras yield pixel-dependent calibration parameters and that this must be accounted for when such cameras are to be employed efficiently for single-molecule localization microscopy. The authors make the point that the thermal parameters also depend on the exposure time and they demonstrate that their calibration applies for a range of exposure times. They proceed to show using super-resolution imaging of biologically-relevant structures that (i) their method allows correction of pixel-dependent artefacts that would otherwise lead to mis-localizations in super-resolution microscopy and (ii) the calibration of also the thermal noise allows the use of industry grade cameras for super-resolution microscopy. The method is implemented in the form of plugins for widely-used pieces of software Micromanager and ImageJ/FIJI, in order to make the method broadly accessible.

Indeed, impact in the field can be made by increasing the accessibility of super-resolution methods by extending the use of industry-grade cameras for localization-based super-resolution microscopy. Additional impact, however, may be limited by the possibility that method does not open up for new opportunities in laboratories where scientific-grade CMOS cameras already are employed in conjunction with state-of-the-art pixel-dependent corrections obtained under the appropriate conditions, e.g. with matching exposure time. To clarify this, I suggest that the authors either add data to their Fig. 2, which show the advantage of the photon-free approach over existing methods (applied appropriately) or present quantitative data that describe the region of the parameter space where the photon-free approach offers something additional.

Along the same lines, I am slightly confused about the spread of the gain values determined in Fig. 1e for the photon-free approach as compared to the conventional method. These data seem to suggest that a pixel-by-pixel calibration based on the photon-free method should be different than (and potentially superior to) the conventional pixel-wise calibration obtained through varying light levels.

I think the authors should explain/demonstrate more carefully how/why the thermally generated electrons obey the same statistical properties as photo-electrons. Do all electrons generated thermally in a given pixel exhibit the same output signal statistics? In the same context, Fig. 1f does seem to contain key data for establishing the appropriateness of the thermally generated photons for use as calibration input, but it is unclear whether the measured pixel statistics are obtained using light exposure or not. If it is done without a light source, then I would suggest that the authors use their photon-free calibration to make predictions for experimental pixel output statistics in various combinations of input light intensity and exposure times. Preferably, this should be done in a manner that reveals the accuracy of the calibration for individual pixels instead of histograms for the occurrences of values.

In general, the manuscript is well written and properly referenced. I do believe, however, that the manuscript could/should use slightly more (in-line) math to communicate the relationships between the different signal quantities and noise sources. For example, "The mean and variance of the signal with no light reaching the camera correspond to offset and read noise squared" (p. 2) could be made first-pass readable by admitting an equation or two.

Additional minor issues:

- Fig. 1c should have color bars to show the values of the various maps to allow easier connection with the panels above.
- The error bars in the legends of Fig. 2k,l should be defined. If they are s.e.m., the shift in the lateral direction is not statistically significant.

REVIEWER COMMENTS

Reviewer #1 (Remarks to the Author):

This manuscript concisely describes a method of pixel-wise calibration for (s)CMOS cameras, which are becoming ubiquitous in fluorescence microscopy including super-resolution applications. The method uses pixel-wise thermal noise and different exposure times to find gain, offset, readnoise and thermal noise. Readnoise and thermal noise can be combined to give a total camera noise, while thermal noise average and the offset can be combined to give a pixel-wise offset. The paper is very complete, even showing the capability with noise CMOS camera. The questions I would have on performance have been addressed in the manuscript.

One small detail I (as a potential user) would like to know is for how long does the calibration remain valid? Although it does seem simple enough to do daily, or just before every long SR data acquisition.

I will likely implement this method in our setup.

We thank Reviewer 1 for the very positive assessment. To address the comment on how long a calibration remains valid, we added data showing the difference in two ACCENT calibration runs on the same camera three years apart (2018 vs. 2021) to the supplementary information.

Reviewer #2 (Remarks to the Author):

In their manuscript entitled "Photon-free (s)CMOS camera characterization for artifact reduction in high- and super-resolution microscopy", Diekmann et al. develop a new (and potentially smart) method for calibration of CMOS cameras of both scientific and industry grade. The novelty of the approach is the fact that thermally generated dark current rather than photonic input is used as the basis for calibration. Consequently, only dark images obtained with the shutter closed at various exposure times are needed, which simplifies data acquisition. It is well-known that CMOS cameras yield pixel-dependent calibration parameters and that this must be accounted for when such cameras are to be employed efficiently for single-molecule localization microscopy. The authors make the point that the thermal parameters also depend on the exposure time and they demonstrate that their calibration applies for a range of exposure times. They proceed to show using super-resolution imaging of biologically-relevant structures that (i) their method allows correction of pixel-dependent artefacts that would otherwise lead to mis-localizations in super-resolution microscopy and (ii) the calibration of also the thermal noise allows the use of industry grade cameras for super-resolution microscopy. The method is implemented in the form of plugins for widely-used pieces of software Micromanager and ImageJ/FIJI, in order to make the method broadly accessible.

Indeed, impact in the field can be made by increasing the accessibility of super-resolution methods by extending the use of industry-grade cameras for localization-based super-resolution microscopy. Additional impact, however, may be limited by the possibility that method does not open up for new opportunities in laboratories where scientific-grade CMOS cameras already are employed in conjunction with state-of-the-art pixel-dependent corrections obtained under the appropriate conditions, e.g. with matching exposure time. To clarify this, I suggest that the authors either add data to their Fig. 2, which show the advantage of the photon-free approach over existing methods (applied appropriately) or present quantitative data that describe the region of the parameter space where the photon-free approach offers something additional.

We also want to thank reviewer 2 for the thorough and positive evaluation. It is correct, that also more standard, but also more elaborate calibration approaches result in proper acquisition

of calibration maps, as long as the exposure time is matched. However, according to our knowledge, many labs do actually not perform such corrections, since they find it too difficult to measure the pixel-dependent effects correctly.

To clarify this, we added the following sentence to the main text “We therefore conclude that our method in determining the relevant camera characteristics is equivalent to the traditional approach, but offers the advantage of full automation and calculation for arbitrary exposure times.”.

Along the same lines, I am slightly confused about the spread of the gain values determined in Fig. 1e for the photon-free approach as compared to the conventional method. These data seem to suggest that a pixel-by-pixel calibration based on the photon-free method should be different than (and potentially superior to) the conventional pixel-wise calibration obtained through varying light levels.

Our gain calibration is performed in the very low signal regime, so the precision on the SINGLE PIXEL LEVEL is less precise than with the conventional approach of using different light levels distributed among the entire dynamic range of the camera. However, the benefit of a pixel-wise gain correction is anyways questionable and even advised against by camera manufacturers. We had multiple discussion over the years with Hamamatsu representatives. Only as an intermediate step, we calculate the gain values on a single pixel level. For the conversion of counts to (photo)electrons, we then use the MEDIAN of all intermediate single pixel gain values. Optionally, we multiply by the flat map, which is of very low variance among pixels, cf. Supplementary Fig. 2d,h.

To clarify this point, we added the following statement to the main text: “Note that our approach operates in the very low signal regime of a few electrons only, and so, the gain estimation on the single pixel level is not very precise. Therefore, we use the median of all single pixel gain values as one global gain value.”

I think the authors should explain/demonstrate more carefully how/why the thermally generated electrons obey the same statistical properties as photo-electrons. Do all electrons generated thermally in a given pixel exhibit the same output signal statistics? In the same context, Fig. 1f does seem to contain key data for establishing the appropriateness of the thermally generated photons for use as calibration input, but it is unclear whether the measured pixel statistics are obtained using light exposure or not.

- *We added Figure 1a to show the equivalence of the signal statistics for (i) thermally generated electrons only, (ii) mixture of thermally generated and photo electrons and (iii) photo electrons only.*
- *We added the statement “from dark frames” to the figure legend to clarify that no light exposure was used here.*

If it is done without a light source, then I would suggest that the authors use their photon-free calibration to make predictions for experimental pixel output statistics in various combinations of input light intensity and exposure times. Preferably, this should be done in a manner that reveals the accuracy of the calibration for individual pixels instead of histograms for the occurrences of values.

In general, the manuscript is well written and properly referenced. I do believe, however, that the manuscript could/should use slightly more (in-line) math to communicate the relationships between the different signal quantities and noise sources. For example, “The mean and variance of the signal with no light reaching the camera correspond to offset and read noise squared” (p. 2) could be made first-pass readable by admitting an equation or two.

We added in-line math to the corresponding paragraph.

Additional minor issues:

- Fig. 1c should have color bars to show the values of the various maps to allow easier connection with the panels above.

We added the color bars.

- The error bars in the legends of Fig. 2k,l should be defined. If they are s.e.m., the shift in the lateral direction is not statistically significant.

We apologize for the confusion: The bars were meant to indicate the maxima of the corresponding double Gaussian fits, but actually looked like error bars. We removed them in the revised manuscript. The numbers $(XX \pm YY)$ nm are calculated by error propagation from the 95% confidence intervals of the parameters in the double Gaussian fits.

REVIEWERS' COMMENTS

Reviewer #1 (Remarks to the Author):

The authors have made updates to the paper and addressed my primary concern. I recommend for publication.

RESPINSE TO REVIEWERS COMMENTS

Reviewer #1 (Remarks to the Author):

The authors have made updates to the paper and addressed my primary concern. I recommend for publication.

We thank Reviewer 1 once again for the positive assessment.